# Pancreatic Enzyme Replacement Therapy in Cystic Fibrosis

**DOI:** 10.3390/nu14071341

**Published:** 2022-03-23

**Authors:** Peter N. Freswick, Elizabeth K. Reid, Maria R. Mascarenhas

**Affiliations:** 1Helen DeVos Children’s Hospital, Grand Rapids, MI 49503, USA; 2Children’s Hospital of Philadelphia, Philadelphia, PA 19104, USA; reide1@chop.edu (E.K.R.); mascarenhas@chop.edu (M.R.M.)

**Keywords:** cystic fibrosis, pancreatic insufficiency, PERT, pancreatic enzymes, nutrition

## Abstract

While typically considered a pulmonary disease, cystic fibrosis patients develop significant nutritional complications and comorbidities, especially those who are pancreatic insufficient. Clinicians must have a high suspicion for cystic fibrosis among patients with clinical symptoms of pancreatic insufficiency, and pancreatic enzymatic replacement therapy (PERT) must be urgently initiated. PERT presents a myriad of considerations for patients and their supporting dieticians and clinicians, including types of administration, therapy failures, and complications.

## 1. Introduction

Cystic fibrosis (CF) is the most common life-shortening autosomal recessive disorder in North America, effecting ~30,000 people in the United States alone [1]. CF is caused by various mutations in the gene that codes for the cystic fibrosis transmembrane conductance regulator (CFTR) gene which encodes a cyclic adenosine monophosphate regulated chloride channel, responsible for chloride and bicarbonate secretion across epithelia cells. Abnormal chloride transport through the CFTR leads to viscous sodium bicarbonate-depleted fluid, and ultimately can lead to pancreatic insufficiency with subsequent malnutrition and malabsorption, fat-soluble vitamin deficiencies, and progressive obstructive lung disease [2].

While many now consider CF primarily a pulmonary disease, CF was initially recognized by Dr Dorothy Andersen in 1938 as a distinct diagnosis for patients with failure to thrive [3]. She labeled the disease “cystic fibrosis of the pancreas” based off her autopsy findings of children who died of malnutrition. As such, the lack of pancreatic function was the initial defining characteristic of CF with many children succumbing to malnutrition before CF lung disease progressed. Now, patients with CF are quickly tested for pancreatic sufficiency and are thus characterized as “pancreatic sufficient” (PS) or “pancreatic insufficient” (PI) [4].

The pancreas consists of the exocrine pancreas that produces pancreatic enzymes and the endocrine pancreas that produces insulin. In this review, the term pancreatic insufficiency refers to inadequate production of pancreatic enzymes, bicarbonate and fluid by the pancreas resulting in maldigestion and malabsorption. Approximately 85% of CF patients have evidence of maldigestion due to pancreatic insufficiency requiring treatment [5]. CF patients experience progressive and substantial pancreatic injury early in life, even in utero, thus often developing PI in infancy [4].

While first recognized in 1938, clinicians had a significant breakthrough in understanding CF disease pathology in 1989 when Lap-Chee Tsui and colleagues cloned the CF gene. From this discovery, scientists have defined six classes of mutations CFTR gene, classes I–VI, stratified in decreasing severity. Such a genotype dichotomy has helped predict a patient’s likelihood of PI: patients with two copies of class IV, V, or VI mutations tend to be PS, whereas those with two copies of class I, II, or III mutations tend to be PI [5].

Aggressive nutritional therapy is paramount to improved clinical outcomes for patients with CF. With the development of comprehensive North American CF care centers, clinicians could recognize varying outcomes between care centers. Corey et al. compared various metrics between Boston and Toronto’s CF care centers in 1988 finding that Toronto’s patients had significantly better survival despite having similar FEV_1_ values. Possibly explaining this drastic difference, patients in Toronto had drastically better weight for height as Toronto advocated for a high-fat, high-calorie diet with aggressive pancreatic enzyme replacement therapy (PERT), whereas Boston advocated for a low-fat, high-calorie diet with less emphasis on PERT [6]. A subsequent prospective study validated Corey et al.’s findings, demonstrating that higher weight percentile in early childhood predicts improved FEV_1_ and survival at 18 years old [7]. Given the advent of the newborn screen (NBS) [8] and a focused interest in therapeutic nutrition and PERT in CF children beginning in infancy, median World Health Organization height percentiles in US children with CF younger than 2 years old increased from less than 20% in 1993 to 44% in 2019 [9]. Leung et al., compared a historic cohort of infants from 1994 to1995 to their prospective cohort concluding that the initiation of a universal NBS, aggressive PERT, and early nutritional therapy improved the nutritional status of CF infants [10].

Given numerous studies demonstrating aggressive nutrition’s vital impact on health in the CF population and the subsequent use of PERT, this review will explore current updates on the clinical presentation of patients with PI CF, the effects of PERT, various nuances to consider, and special considerations of PERT therapy in the age of CFTR modulators.

## 2. Clinical Considerations

### 2.1. Presentation

Patients with malabsorption present classically with malodorous oily stools that can be difficult to flush, chronic diarrhea, failure to thrive, weight loss, bloating, and dyspepsia [11]. All causes of malabsorption, not just pancreatic insufficiency, will present similarly, and thus a broad differential diagnosis should be considered when approaching an infant or child with malabsorption symptoms [4,11,12] (see Table 1). Regardless, anytime a pediatric patient presents with failure to thrive, a full differential diagnosis must be considered.

Before universal newborn screening was available in the USA, cystic fibrosis would present with meconium ileus, rectal prolapse, failure to thrive and weight loss, respiratory difficulties, and steatorrhea, among others [2]. As of 2010, all 50 states are performing CF screening at birth [13]. Screening typically entails measuring an infant’s immune-reactive trypsinogen (IRT) serum level, occasionally coupled with DNA analysis for common CFTR mutations. Should an infant screen positive, he/she should be immediately referred for a confirmation sweat chloride test [14]. However, some infants with CF will not screen positive [15,16], thus clinicians must always remain vigilant to the diagnosis of CF in all infants with concerning symptoms.

### 2.2. Diagnosis

Since newborn screening has been in place, infants often present for the initial CF center visit at 2 weeks of age. For infants with identified CFTR mutations associated with PI or poor initial weight gain, PERT should be initiated as soon as possible, even before a patient has been officially diagnosed as PI. Caregivers may bring a stool sample to be sent for fecal elastase (FE) and PERT administration may be demonstrated and started at the initial appointment. Taking PERT will not affect a FE test result, and PERT can be discontinued if the test result is negative for PI.

Early diagnosis of PI and intervention leads to improved nutritional status [10], regardless of age [4]. PI testing can prove difficult, as testing only easily detects severe PI and there is no gold standard for PI diagnosis or severity [17]. PI testing is either considered indirect or direct.

Indirect testing evaluates the secondary fecal effects due to lack of pancreatic enzymes [17]. The historical gold standard has been the 72 h fecal fat test, with a coefficient of fat absorption normal if ≥85% if the patient is less than 6 months of age and ≥93% if 6 months or older [18]. However, 72 h tests are unpleasant and time consuming. An acid steatocrit (AS) test has been proposed as an alternative test. Walkowiak et al., compared the AS among CF patient without or with mild steatorrhea, but unfortunately found that AS did not reflect the fecal fat excretion in these CF patients [19]. However, a FE > 100 micrograms/gram stool has a 99% negative predictive value for pancreatic insufficiency. FE is easier to obtain than a 72 h fecal fat and, as proven to be a valid screening method for PI, has become the standard test for PI [20]. In general, indirect tests are only accurate for advanced stages of PI [17].

Direct tests measure the secreted pancreatic enzymes and bicarbonate [17]. Cholecystokinin stimulates pancreatic enzyme secretion and secretin stimulates pancreatic bicarbonate secretion. Both Dreiling tubes (nasal tube with gastric and duodenal ports to decompress the stomach and measure duodenal secretions, respectively) and endoscopy have been utilized for direct exocrine pancreatic testing [4]. However, among patients with chronic pancreatitis, direct PI testing was similar between endoscopic and Dreiling tube methods, and the endoscopic method eases the performance of these tests [21].

### 2.3. Subsequent Testing

After the diagnosis of PI in a patient with CF, it is important to note that the fecal elastase values may fluctuate significantly in the first year of life. O’Sullivan et al. following 61 CF patients obtained at least 8 fecal elastase levels initially obtained at age < 3.5 months and the final level obtained at age ≥ 9 months old. They found significant variability in fecal elastase testing across all initial fecal elastase groups. In total, 27% of those with an initial fecal elastase between 50–200 micrograms/gram had at least one fecal elastase > 200 micrograms/gram. Furthermore, 15% of those with an initial fecal elastase > 200 micrograms/gram had a fecal elastase <200 micrograms/gram by the end of the first year [22].

Given this variability, clinicians should always consider the evolution of a patient’s pancreatic exocrine function. Thus, guidelines now recommend that PS CF children and adults undergo an annual assessment of pancreatic function by fecal elastase measurement, and more frequent tests are needed for poor growth or inadequate nutritional status [23].

## 3. Treatment

### 3.1. Types of PERT

PI is treated with PERT capsules which contain pancreatic extract (lipase, protease, and amylase) to replace the missing endogenous pancreatic enzymes. All the current FDA-approved products are derived from porcine origin. Patients who follow religious or cultural preference to avoid pork products may be given a medical dispensation. Non-porcine enzymes are present in clinical trials; however, there is no effective alternative currently available. There are over the counter preparations available which are sold as digestive aids. These are only available in very low doses and are not efficacious to treat PI.

There are three formulations: enteric-coated, non-enteric coated, and a lipase enzyme cartridge. There are different brands of PERT which come in multiple sizes and dosing strengths based on lipase units (See Figure 1).

Products for oral use are generally in capsule form containing microspheres or microtablets with a pH-sensitive coating protecting the enzymes from gastric acid and allow activation in the more alkaline environment of the duodenum.

### 3.2. Oral PERT Administration

The capsules are administered by either swallowing whole or opening the capsules and sprinkling the contents in a small amount of acidic food such as applesauce. The enzymes beads or microtablets must be swallowed whole; they should not be chewed or crushed.

PERT may be administered to infants by sprinkling the capsule contents in a small amount of applesauce and offering right before breast or bottle feeding. The infant’s mouth should be checked for retained beads and mucosal irritation. Starting a skin barrier cream is recommended because some of the beads may pass through the immature intestine and cause perianal irritation.

PERT dose can be determined by patient weight or by the fat content of the meal or snack. PERT dosing should be individualized (see Table 2); PI CF patients should be started on the lowest effective dose and then have the dose titrated based on weight gain and gastrointestinal symptoms to the lowest effective dose, not to exceed 2500 lipase units/kg/meal and a total of 10,000 lipase units/kg/day. The number after an enzyme name denotes lipase units per capsule multiplied by 1000. Several companies provide varying capsule dosages with varying bead sizes, best illustrated by the excellent image prepared by Karen Maguiness, MS, RD and used with permission (see Figure 1). Some young infants may transiently exceed this upper limit of dosing recommendations due to the need for frequent PERT dosing with on demand feeding and should be monitored by an experienced clinician [24]. Optimal enzyme dosing is associated with typical rates of weight gain and growth determined by genetic potential and evidence of some improvement of indirect markers of maldigestion such as vitamins levels. Doses are adjusted based on clinical symptoms of poor weight gain and/or fat malabsorption by increasing 1 capsule per dose not to exceed 2500 lipase units/kg/meal. Another consideration is tailoring the enzyme dose to the specific foods by dosing per fat gram. Always consider consultation with a nutrition or PERT expert when dosing enzymes to determine the best method for use and for dose titration recommendations.

### 3.3. PERT Administration for Bolus and Continuous Enteral Feeding

Providing PERT for patients with PI receiving continuous enteral feeding has been challenging. Until recently, a common practice has been to provide oral PERT capsules at the beginning and end of the tube feeding. This off label practice is not practical from a physiologic sense to provide a bolus of PERT which lasted 45–60 min during a continuous infusion of nutrients often over 4–8 h or more. Some but not all patients tolerate this method and achieve desired weight gain. However, some may have ongoing abdominal pain, bloating, bowel urgency and other undesirable side effects and often results in decreased patient adherence.

The PERT dose for bolus enteral feeding can be based on the weight, or more accurately determined based on the grams of fat in the formula. A typical dose would be calculated at approximately 2000 lipase units per gram of fat in the formula, with a range between 500–4000 lipase units/gram of fat (see Table 2 PERT dosing recommendations). PERT may be administered orally at the start of the bolus feed with enteric-coated capsules either swallowed whole or opened and sprinkled on acidic fruit sauce. For instances when PERT cannot be swallowed, a non-enteric-coated tablet may be carefully crushed and added to the volume of formula for the bolus. The dose would be based on the grams of fat in the formula, typically at 2000 lipase units per gram of fat and rounded to the nearest ½ tablet size within a range of 500–4000 lipase units per gram of fat (see Table 2 PERT dosing recommendations).

The first FDA-approved device for patients with fat malabsorption ages 5 years and older receiving continuous enteral feeding is a lipase only cartridge (Relizorb) placed in-line with the tube through which the enteral feeding flows. The lipase inside the cartridge continuously breaks down long-chain triglycerides into absorbable components throughout the entire duration of the enteral feeding which then flow into the patient. One cartridge is recommended for 500 mL of formula at flow rates between 10–120 mL per hour. Cartridges can be connected in tandem to accommodate volumes between 500–1000 mL at rates between 24–120 mL per hour (see Table 3). Several studies report good results with improved weight gain, improved essential fatty acid profiles, and fat-soluble vitamins as well as improved gastrointestinal symptoms [26,27]. There is a list of compatible formulas on the company website with corresponding fat hydrolysis data which are updated as more information becomes available (relizorb.com). Note there are no data for infant formulas or breast milk as the cartridge has not been FDA approved in these age groups although there may be case reports.

Another off-label method used for pancrelipase coverage during continuous enteral feeding when unable to use the Relizorb cartridge is Viokace, a non-enteric-coated pancrelipase tablet which must be crushed and added to the formula before infusion (see Table 3). This method is preferable to giving oral enteric-coated pancrelipase before overnight formula infusion and in cases when the Relizorb cartridge is not compatible with the prescribed enteral regimen. The Viokace tablets are crushed carefully to a fine powder and added to the formula prior to infusion. Care must be taken to protect eyes and skin and not breathe in the powder. Dosing is based on grams of fat in the formula or breast milk (see Table 2).

### 3.4. Considerations for Treatment Failure

Poor growth is common among the CF population, even in clinics with robust dietician support. Numerous considerations must be entertained. A primary cause of poor growth is poor caloric intake, especially common given the many symptoms that CF patients have (abdominal distension, abdominal pain, constipation, etc.). Of course, other causes of malabsorption and other factors besides PI that may affect fat malabsorption and/or weight loss must be considered as well, such as constipation or celiac disease (see Table 1 for a more complete differential diagnosis). Among CF patients with mild PI CF-associated liver disease, Drzymala-Czyz et al., found that ursodeoxycholic acid supplementation enhanced fat absorption [28].

Enzyme failure must also be carefully considered; for common causes of enzyme failure see Table 4. As CF patients have increased gastrointestinal acidity [29], and acidity may decrease enzyme activity efficacy [30] and/or precipitate bile acids in the CF intestine [31] (and thus decreased micelle formation and fat absorption), adding acid-suppressing agents may improve fat absorption [32]. However, acid-suppression therapies (most commonly proton pump inhibitors) are still being evaluated and other adjustments might be considered first. In general, an acid suppression therapy trial is typically limited with clear endpoints such as weight gain [4].

### 3.5. PERT Complications

Despite PERT’s vital component to any PI CF patient treatment regimen, PERT has side effects that must be considered by the clinician. Fibrosing colonopathy is a well-known complication of PERT associated with high PERT doses [33]. FitzSimmons et al.’s case-controlled study from 1997 demonstrated that higher doses of PERT were associated with fibrosing colonopathy, finding that affected patients had a daily dose of PERT 2.5 times higher than the unaffected control group [34]. However, even the control group had an daily average PERT intake of 18,000 units lipase/kg/day [34], far higher than the current US care guidelines [18]. Thus, it has been argued that the maximal PERT dose of 10,000 units lipase/kg/day is not evidence based, and clinicians can consider higher doses especially in infants as patient’s needs necessitate [33].

## 4. PERT Efficacy

While PERT efficacy could be measured by weight changes over time and PI symptom improvement, these endpoints are lagging and subjective. Thus, nearly all studies for PERT efficacy utilize coefficient of fat absorption (CFA) over 72 h to measure improvement in steatorrhea, and coefficient of nitrogen absorption (CNA) to quantify changes in azotorrhea. While CFA is an especially accurate and precise measurement for steatorrhea in otherwise healthy individuals, CFA is less accurate and precise among patients with CF. However, CFA remains the primary outcome measure to investigate PERT efficacy [35].

Accurate measurement of fecal fat is not only important for diagnosing PI, but also PERT dosing. Overall, 72 h fecal collections for CFA are arduous and difficult for patients. Caras et al. demonstrated that the mean calculated CFA of three random stool samples over 3 days was as sensitive to predict a percentage fat < 30% as a 72 h CFA [36]. While still difficult to obtain, these random stool samples would be more easily obtained than a strict 72 h fecal collection. Clinically, random stool collections are not used to titrate PERT dosing.

A recent metanalysis on PERT’s efficacy in patients with chronic pancreatitis showed that PERT significantly improved CFA and CNA, and among randomized controlled trials PERT improved GI symptoms and decreased fecal weight, fecal fat, and nitrogen excretion [37]. A Cochrane review recently published noted that there are no high-quality trials comparing PERT to placebo in individuals with CF. However, when comparing enteric-coated microspheres (ECM) to enteric-coated tablets (ECT), ECM has superior outcomes regarding abdominal pain, stool frequency, and fecal fat excretion suggesting ECM’s significant efficacy among patients with CF. Interestingly, there are no outcome differences among the various formulations of ECMs [38].

## 5. PERT and Highly Effective Modulator Therapies

Severe PI has been thought to be irreversible, and CF-related PI is often present from birth or develops in most instances within the first year of life. The introduction of highly effective modulator therapies (HEMTs) has demonstrated that pancreatic function is more dynamic than previously thought. HEMTs include ivacaftor (IVA) for gating mutations such as G551D and IVA/tezacaftor/elexacaftor for at least a heterozygous F508del mutation [39]. In some cases, long-term use of HEMTs (especially IVA for gating mutations) may reverse the PI in some PI CF patients as seen in the following studies.

In the ARRIVAL study, 19 children aged 12 to <24 months with at least one gating mutation were placed on CFTR modulator IVA for 24 weeks [40]. After the 24-week trial, the patient’s fecal elastase improved from a mean of 182.2 ug/g to 326.9 ug/g and immunoreactive trypsinogen improved from a mean of 1154.9 ng/mL to 505.4 ng/mL. Eleven children were considered pancreatic insufficient at baseline (all 11 had baseline values < 50 μg/g). Nine of these children had both baseline and week-24 fecal elastase values; six of these nine children had fecal elastase > 200 μg/g at week 24, a value consistent with normal pancreatic function.

In the 24-week KIWI trial of IVA given to older children ages 2–5 years, a mean increase in FE of 99.8 ug/g was observed and sustained throughout the open-label extension KLIMB study. Subjects were followed from week 24 to 84 although no further significant changes were noted in FE. For the 18 subjects who had FE measurements completed at baseline and week 84, one had a FE > 200 ug/g at baseline and five had FE > 200 ug/g at week 84 [41].

Thus, it seems that HEMTs can improve, and possibly rescue, pancreatic exocrine function in some patients, even in older patients according to these studies. There is significant variability in fecal elastase response among the patients in the KIWI and ARRIVAL trials, suggesting that a subset of CF PI patients may have enough residual acinar activity that can be rescued. Given this variability, approaches to PERT therapy for patients on HEMT are currently highly variable and not standardized. As of this time, we are in the early days of HEMT use and have yet to fully appreciate the effects of long-term therapy on exocrine pancreatic function.

## 6. Conclusions

The CF population presents unique considerations for their physicians and dieticians caring for them. Improved nutrition for the CF patient is paramount for improved clinical outcomes and survivability, and PERT is essential for PI CF patients. We have explored the nuances, considerations, administrative techniques, and complications of PERT in this review. As PI CF patients have unique disease pathology, the providers and dieticians that care for CF patients must have a robust knowledge of PERT.

## Figures and Tables

**Figure 1 nutrients-14-01341-f001:**
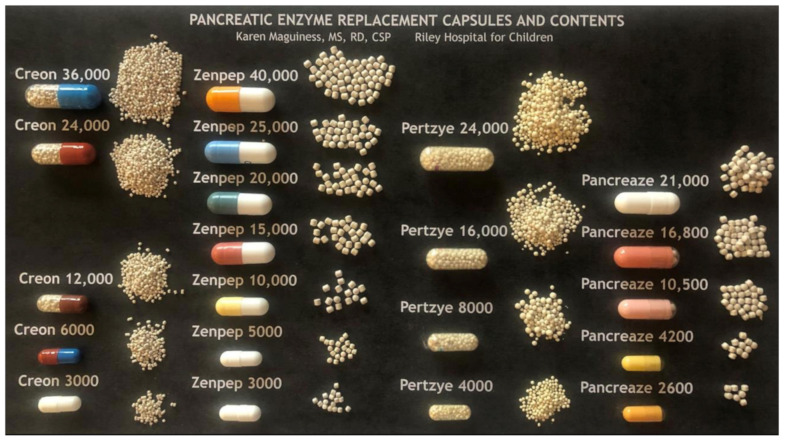
PERT Capsules and Contents. Reprinted with permission from Karen Maguiness, MS, RD, CSP, Riley Hospital for Children.

**Table 1 nutrients-14-01341-t001:** Causes of childhood malabsorption.

Intestinal	Extra-Intestinal
Crohn’s disease Celiac disease Small intestinal bacterial overgrowth (SIBO) Infectious diarrhea (*Giardia, Cryptosporidium*) Brush border enzyme deficiencies Short bowel syndrome Acrodermatitis enteropathica	Cystic fibrosis Zollinger-Ellison syndrome Gastroparesis Chronic cholestasis Schwachman-Diamond syndrome Johanson-Blizzard syndrome Pearson syndrome Jeune syndrome Pancreatic aplasia Cholestatic liver disease

**Table 2 nutrients-14-01341-t002:** PERT Dosing Recommendations.

Patient Age	Age-Based Dosing Recommendations	Focus Guidance About Administration	Titration
PERT for Oral Feeds—Use Enteric-Coated Formulation			
Premature and full-term infants, <12 months	Initiate when taking >60 mL per feed (formula/breastmilk). Starting dose: 3000 lipase units/feed Range: 1000–2500 lipase units/kg/feed Max: 10,000 lipase units/kg/day	Open capsule, sprinkle enzyme beads on a small amount of applesauce Administer at start of feed Give by mouth even if portion of feed is enteral Never give via feeding tube (clogs tube) Check infant’s mouth for retained beads and mucosal irritation Start skin barrier cream and monitor for perianal irritation	Increase by 1 capsule per dose based on clinical symptoms of malabsorption and/or poor weight gain Max dose may transiently exceed 10,000 lipase units per kg/day due to frequency of infant feedings
Children and Adolescents	Starting dose 1–4 years 1000 lipase units/kg/meal Titrate to max 2500 lipase units/kg/meal ≥4 years 500 lipase units/kg/meal Titrate to max 2500 lipase units/kg/meal Range: 500–2500 units/kg/meal Max: 10,000 lipase units/kg/day Snack dose: Half of the meal dose	Give capsule by mouth Give by mouth even if portion of feed is enteral If unable to swallow capsule: open capsule, sprinkle enzyme beads on a small amount of applesauce Meals lasting longer than 30 min: split dose and administer halfway through meals	Increase by 1 capsule per dose based on clinical symptoms of malabsorption and/or poor weight gain. Consult CF RD or another PERT expert and dose as needed
**PERT for Tube Feeding—Bolus Feeds**			
Enteric-Coated Enzyme	Weight based Starting dose Children: 500–1000 lipase units/kg/feed Range: 500–2500 lipase units/kg/feed Max: 10,000 lipase units/kg/day	Only use if patient is able to take enzymes by mouth Give at start of feed Give at start of feed Start at lower end of dosing range Initiate when taking ≥60 mL per feed Never crush or chew enzymes	
	Grams of fat based Typical dose: 1800–2200 lipase units/g of fat Range: 500–4000 lipase units/g of fat	Dose enzymes based on total grams of fat in the formula per RD recommendations Non-CF: start at lower end of dosing range	
Non-Enteric-Coated Enzyme (Viokace^®^)	Grams of fat based Typical dose: 1800–2200 lipase units/g of fat Range: 500–4000 lipase units/g of fat	Use ONLY if patient is unable to take enteric-coated enzymes by mouth Crush Viokace and add to formula Lipase units that come from Viokace do NOT count towards total max dose per day of 10,000 units lipase/kg/day Round to the nearest ½ tablet of Viokace. Dosage options are either 10,440 or 20,880 lipase units per tablet Non-CF: start at lower end of dosing range	
**PERT for Tube Feeding—Continuous/Overnight Feeds**			
RELiZORB^TM^	Patient is ≥5 years of age Starting dose: 1 cartridge per 500 mL formula Max: 2 cartridges/24 h period Optimal rate of feed: 24–120 mL/h	Preferred method for continuous tube feeds Avoid fiber containing and blenderized formulas, can clog cartridge Minimum tube feeding rate is 24 mL/h If rate is lower than 24 mL/h, use crushed (Viokace) method below	
Non-Enteric-Coated Enzyme (Viokace^®^)	Grams of fat based Typical dose: 1800–2200 lipase units/g of fat Range: 500–4000 lipase units/g of fat	Crush Viokace and add to formula Initiate when taking ≥15 mL per hour (formula/breast milk)If unable to use RELiZORB, this method provides the next best option Lipase units that come from Viokace do NOT count towards total max dose per day of 10,000 units lipase/kg/day Round to the nearest ½ tablet of Viokace. Dosage options are either 10,440 or 20,880 lipase units per tablet Non-CF: start at lower end of dosing range	
**Enteric-Coated Enzyme**	Given orally only Dose based on weight or grams of fat in tube feeding formula Weight based Range: 500–2500 lipase units/kg/feed Max: 10,000 lipase units/kg/day Grams of fat based Typical dose: 1800–2200 lipase units/g of fat Range: 500–4000 lipase units/g of fat	This option is for patients >2 years on overnight feeds who can take oral enzymes when RELiZORB or Viokace are not available Use a meal dose of enzymes at start of tube feeds. Patients may need an additional half dose at the end of the feed Do not use for 24 h continuous feeds Initiate when receiving >15 mL per hour (formula/breast milk) Do not recommend enteric-coated enzymes through any enteral tube due to risk for clogging Non-CF: start at lower end of dosing range	

Copied with permission from Children’s Hospital of Philadelphia’s Clinical Pathway for PERT in Children with or at Risk for Exocrine PI [25].

**Table 3 nutrients-14-01341-t003:** Enzyme Types and Considerations.

Enzyme Type	Considerations
Enteric-Coated	Capsules containing enteric-coated beads or microtablets Coating protects enzymes from gastric acid, allows activation in duodenum Administration: Give by mouth at the start of feeds/meals/snacks/beverages. ○ Swallowed whole or opened, and contents sprinkled on a small amount of applesauce. ○ Give even if portion of their feeding is via an enteral tube. Do NOT: ○ Administer via feeding tube, will clog tube. ○ Crush or chew enzyme beads.
Non-Enteric-Coated (Viokace^®^)	Powdered tablets Most often used for patients on tube feedings Administration: Crush and add to enteral formula to pre-digest nutrients in feeding bag prior to enteral tube administration. Do NOT give orally
Lipase Cartridge (RELiZORB^TM^)	Enzyme cartridge only containing enzyme lipase For patients on continuous tube feeds only Administration: Cartridge is connected in-line with the enteral tube feeding set. Enteral formula flows through the cartridge and fat is digested in the formula. Refer to RELiZORB^TM^ manufacturer’s data sheet for compatible formulas

Copied with permission from Children’s Hospital of Philadelphia’s Clinical Pathway for PERT in Children with or at Risk for Exocrine PI [25].

**Table 4 nutrients-14-01341-t004:** Troubleshooting PERT Failure.

PERT Considerations	
Timing of PERT administration	Enzymes should be administered prior to eating all meals and snacks. ○ Food should be eating within 45–60 min of enzyme dose to ensure appropriate enzyme activity. Slow eaters, gastroparesis, or fat eaten at end of meal. ○ Consider splitting dose, taking ½ at start of meal and ½ dose partway through the meal. If most food is eaten in 1 sitting. ○ Dose adjustment may be difficult. Consider spreading food and fat intake over course of day
Type of food eaten	Match enzyme dose to food eaten. ○ Consider dosing enzymes based on fat grams.
Storage	Keep lid tightly closed on enzyme container. Enzymes should be kept at room temperature (59–86 degrees Fahrenheit). ○ Heat destroys enzymes: do not store on top of refrigerator or toaster oven, keep out of hot cars. ○ Cold temperatures can harm enzymes, do not refrigerate.
Expiration date	Enzymes degrade overtime so always check the expiration date.

## Data Availability

Not applicable.

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
