# Peer review of "Pancreatic Enzyme Replacement Therapy in Cystic Fibrosis"

_nutrients, 2022, doi:10.3390/nu14071341_

Round 1

Reviewer 1 Report

It is a high quality article concerning nutritional complications in patients with cystic fibrosis. Pancreatic insufficienty can not be neglected in clinical assessment of a patient with cystic fibrosis. The article seems to be a good teaching aid especially for doctors specializing in pediatric diseases. I have only one concern: why authors several times write 'in this chapter'. Was is supposed to be a book chapter or an article?

Author Response

We appreciate the feedback! You are right about the chapter / article confusion. This was initially to be written as a book chapter, and then over time it evolved into a review article appearing in a special section on CF. I have changed all references to "chapter" to say "review". Let me know if you have noticed anything else. Thank you!

Reviewer 2 Report

Manuscript ID: nutrients-1631779
Type of manuscript: Review
Title: Pancreatic Enzyme Replacement Therapy in Cystic Fibrosis.

This is a very interesting and important paper considering various aspects of the treatment of pancreatic insufficiency in patients with cystic fibrosis.

Overall, the manuscript is well written and addresses key topics such as diagnosis, types of treatment, and complications of treatment, which is especially important for clinicians and patients using enzyme replacement therapy. I am not a clinician, only a biologist, so I think this review needs input from such a person who has experience with pancreatic enzyme replacement therapy in cystic fibrosis.

What I feel is missing here in the introduction is a more detailed description of what pancreatic pathology looks like in cystic fibrosis. The authors use the term “pancreatic insufficiency”, however, it would be good to include a few sentences about what exactly it is, what the mechanism is, and what it results in.

Author Response

Thank you for your feedback! This was clearly an oversight on our part. I have added a new paragraph (now the 3rd paragraph), where pancreatic insufficiency is clearly defined. I also discuss the timing of its development, and the mechanism by which CF patients become PI. Let me know if you have any other feedback. Much appreciated! 

Reviewer 3 Report

The manuscript is very interesting and well summarizes the current state of knowledge on pancreatic enzymatic replacement therapy.
I propose to correct/refill only 2 paragraphs:

verse 99 - it is worth specifying what intermediate tests the author has in mind - only the manuscript from the year 2009 was cited, in my opinion, it is worth describing and quoting as well article by Walkowiak et al. (2010) who describes the use of the acid steatocrit indicator [DOI: 10.1002/ppul.21149].

lines 218-225 - among the causes of treatment failure, it is worth mentioning coexisting liver disease and citing the work of Drzymala-Czyz, who mentions liver involvement as the often-overlooked disturbing factor on digestion and absorption of fats in CF [DOI: 10.1097/MEG.0000000000000593].

Author Response

Thank you very much for your feedback! Both of your suggestions are excellent. 

  1. In lines 100-109, I mention the acid steatocrit test citing Walkowiak et al.
  2. In table 1, I had already mentioned that cholestatic liver disease can affect fat absorption, however you have an excellent point that we should specifically mention CFLD and potential treatment with Urso. This is now mentioned in lines 235-27. 

We appreciate your feedback, and let me know if you have any other suggestion. Thank you!